# DEEP SPIKE DECODER (DSD)

## ABSTRACT

Spike-sorting is of central importance for neuroscience research. We introduce a novel spike-sorting method comprising a deep autoencoder trained end-to-end with a biophysical generative model, biophysically motivated priors, and a self-supervised loss function to training a deep autoencoder. The encoder infers the action potential event times for each source, while the decoder parameters represent each source's spatiotemporal response waveform. We evaluate this approach in the context of real and synthetic multi-channel surface electromyography (sEMG) data, a noisy superposition of motor unit action potentials (MUAPs). Relative to an established spike-sorting method, this autoencoder-based approach shows superior recovery of source waveforms and event times. Moreover, the biophysical nature of the loss functions facilitates interpretability and hyperparameter tuning. Overall, these results demonstrate the efficacy and motivate further development of self-supervised spike sorting techniques.

## 1 INTRODUCTION

Spike sorting focuses on analyzing electrophysiological data, detecting occurrences of action potentials, and assigning them to distinct neurons. The resulting spikes-based representation is regarded as a complete, high-resolution, noise-free, and disentangled representation of the underlying neural activity. Spike sorting has been studied extensively for use with extracellular electrode (reviewed by Rey et al. 2015; Carlson & Carin 2019; Hennig et al. 2019) and surface electromyography (sEMG; Farina et al. 2004; Holobar & Zazula 2007; Negro et al. 2016). In this paper, we propose a novel solution to the spike sorting problem that marries biophysical inductive biases and constraints with the power of self-supervision.

Our approach uses a deep autoencoder architecture where the encoder functions as spike sorter and the decoder implements a generative model. The entire model is trained end-to-end in a completely self-supervised fashion by leveraging a set of loss terms that enforce biophysically-motivated priors. In this paper, the proposed approach evaluated using both real-world and realistically simulated sEMG datasets. The results demonstrate the validity of our approach and motivate further development of the overall paradigm of unsupervised deep spike sorting.

## 2 PROBLEM FORMULATION

Here, we briefly discuss the problem formulation, and generative model giving rise to the measured sEMG signal (Merletti & Farina, 2016).

A motor neuron innervates a set of muscle fibers, and together they are referred to as a motor unit (MU). As an MU emits a spike, denoted by the variable $s$, it passes through layers of connective tissue, blood, and skin, before being picked up by the electrodes on the surface. This pattern is referred to as the motor-unit-action-potential, (MUAP). The received sEMG signal at the electrode, is the result of an MU spike being convolved with a kernel, given by the tissue between the $m$'th MU and the $e$'th electrode, $\mathbf{h}_{e,m}$. The result of this convolution is the MUAP-waveform, which - since the spike is binary - is also the shape of the convolution kernel. The kernel shape is directly and physically related to the space between the MU and the electrode.

As other spikes from the same MU are transmitted, they are also convolved with the same kernel $\mathbf{h}_{e,m}$ on the way to the electrode. Therefore, the signal that the $e$'th electrode sees due to the $m$'th MU's spike train $s_m[t]$, is the convolution of the two entities, given by:

$$x_e[t] = \sum_{l=0}^{L-1} h_{e,m}[l]s_m[t-l] + n_e[t] \tag{1}$$

where $n_e[t]$ is the thermal noise at the $e$'th electrode. Of course, there are more than one MUs - in fact there are $M$ of them, so the total result of what the $e$'th electrode sees is a sum of all their effects, and so we have:

$$x_e[t] = \sum_{m=0}^{M-1} \sum_{l=0}^{L-1} h_{e,m}[l]s_m[t-l] + n_e[t] \tag{2}$$

## 3 METHOD

The spike sorting problem can be cast as an unsupervised representation learning problem (Goodfellow et al., 2016; Bengio et al., 2013), where the multi-channel sEMG signal is represented as individual MU spike trains and the corresponding MUAP spatiotemporal waveforms. Here, we tackle this problem using a deep autoencoder where the encoder extracts spike pulse trains and the decoder reconstructs the sEMG signal according to the biophysics of the problem. This network is then trained using loss terms that enforce the reconstruction of the sEMG signal, the sparsity of the spike trains, and the parsimony of the MUAPs, the uniqueness of motor units, and the refractory period of motor neurons.

### 3.1 MODEL ARCHITECTURE

The DSD model can be put into the following form where Equations 3 and 4 represent the encoder and the decoder parts correspondingly. The encoder $f_e(.)$ takes in the sEMG data $x \in \mathcal{R}^{ExT}$ where $E$ is the number of electrodes and $T$ is the number of timesteps, and the encoder weights $w_e$, to infer the spike probabilities $\hat{s} \in \mathcal{R}^{NxT}$ where $N$ is the maximum number of MUs the network is allowed learn. In this representation, individual MU spike probabilities are inferred for all MUs for all timesteps.

$$\hat{s} = f_e(x; w_e) \tag{3}$$
$$\hat{x} = f_d(\hat{s}; w_d) \tag{4}$$

The decoder $f_d(.)$ then takes in the inferred MU spike probabilities $\hat{s}$ and infer the reconstructed sEMG signal $\hat{x}$ using the decoder weights $w_d$. Since the decoder mimics the biophysical model closely, we can recover the MUAP waveforms corresponding to each MU from the decoder weights $w_d$.

Here, the encoder model corresponds to the learned deep spike sorter and the decoder model corresponds to the learned biophysical model. During inference time, we typically run only the learned encoder for spike sorting purposes.

**DSD Encoder**  The encoder function $f_e(.)$ constitutes a deep convolutional neural network (CNN) which is tasked with solving the spike sorting problem. While traditional spike sorting approaches first identify peaks and then cluster them into motor units, our deep spike sorter solves both problems simultaneously in a data-driven manner.

The encoder is implemented as a segmentation network followed by a classifier head. The segmentation network takes in the sEMG data $x \in \mathcal{R}^{ExT}$ and encodes it in a latent space of size $(L, E, T)$ where $L$ is the dimensionality of the latent space. The classifier head then takes in this latent representation and calculates spike scores using a linear classifier, with input size $(L, E)$, for each motor unit that runs through each timestep, which are then passed through a Gumbel-softmax function (Jang et al., 2016) for spike binarization.

We have two variants of the DSD model based on the segmentation network used. In one variant, we use a densely connected neural network architecture (i.e. DenseNets) similar to Huang et al. (2017).

The other variant utilizes skip connections between dense blocks and is based on the one hundred layer tiramisu architecture (Jégou et al., 2017).

**DSD Decoder**   The decoder function $f_d(.)$ is strictly a linear model that mimics the convolutive multi-channel sEMG mixing model. This biophysical model, described in detail by Merletti & Farina (2016), can be described as a single convolution operation where the convolutional kernels are the spatiotemporal waveforms corresponding to each motor unit action potential (MUAP). Using this biophysical model helps us constrain the problem such that, together with the training losses, it coincides with the spike sorting problem. And the interpretability of it allows us to easily recover the MUAP waveforms corresponding to each MU.

In our implementation, we used the canonical tensor decomposition trick (Kolda & Bader, 2009; Hitchcock, 1927), alongside the spatial and the temporal dimensions to significantly reduce the number of learned parameters while maintaining the expressive power necessary to span the space of possible MUAP waveforms.

## 3.2   TRAINING LOSSES

**Reconstruction Loss**   The reconstruction loss is given in Equation 5. It penalizes for the $L_4$ norm of the reconstruction error.

$$L_{\text{reconstruction}} = \|x - \hat{x}\|_4 \tag{5}$$

**Sparsity Loss**   MU spikes can be rare events relative to typical sampling rates of sEMG devices. We use the $L_1$ norm of the spike probabilities to enforce this prior knowledge, as shown in Equation 6.

$$L_{\text{sparsity}} = \|\hat{s}\|_1 \tag{6}$$

**Parsimony Loss**   One of the main challenges in using a deep autoencoder for spike sorting is the cardinality problem. Since we do not know the exact number of motor units, we overparameterize the network based on an upper limit. Then, by penalizing for the proposal of new MUAP waveforms, we are able to force the network to be parsimonious with respect to the set of MUAP proposals it uses to explain a given sEMG signal. We use the $L_1$-$L_2$ norm to enforce this group sparsity as in Bach et al. (2012).

$$L_{\text{parsimony}} = \sum_{g \in \mathcal{G}} \left\| \frac{\partial w_d^g}{\partial t} \right\|_2 \tag{7}$$

Equation 7 describes the parsimony loss. The partitioning for $L_1$-$L_2$ norm is defined such that each partition, $g \in \mathcal{G}$, corresponds to a tensor $w_d^g$ that describes the local spatiotemporal waveform around a neighborhood of electrodes for a particular MU. This forces the network to minimize both the number of MUAPs and their spatial footprints. Also note that we apply this loss not directly on the spatiotemporal waveforms, but rather on their first derivatives with respect to time, in order to enforce temporal smoothness.

**Uniqueness Loss**   The uniqueness loss forces the MUAP waveform proposals to focus on explaining different phenomena. It implements a joint prior on the set of MUAP waveforms and their corresponding MU spike trains. This is done by penalizing for the joint occurrence of the temporal similarity between MU spike trains and the spatial similarity between their MUAP waveforms.

$$L_{\text{uniqueness}} = \left\| \text{vec} \left( \left( \tilde{s}\tilde{s}^T \right) \odot \left( mm^T \right) \odot (J - I) \right) \right\|_1 \tag{8}$$

In Equation 8, the terms $\tilde{s}$ and $m$ correspond to a dilated version of the spike trains and the spatial energy distribution of the MUAP waveforms. The matrices $J$ and $I$ correspond to an all-ones matrix and an identity matrix respectively. The operators $\text{vec}(.)$ and $\odot$ correspond to vectorization operation and the hadamard element-wise product.

**Refractory Period Loss** Refractory period of motor units describe the amount of time needed between two spikes from a single motor unit. The refractory period loss encodes this prior knowledge by penalizing for multiple spike occurences from a single MU within its time window. It can be implemented efficiently as in Equation 9.

$$L_{\text{refractory}} = \|\text{ReLU}\left(\text{sumpool}\left(s\right) - 1\right)\|_1 \tag{9}$$

**Total Loss** The total loss is a combination of the above loss items as shown in Equation 10 where $\alpha$, $\beta$, $\gamma$, and $\eta$ are hyper-parameters to be tuned.

$$L_{\text{total}} = L_{\text{reconstruction}} + \alpha L_{\text{sparsity}} + \beta L_{\text{parsimony}} + \gamma L_{\text{uniqueness}} + \eta L_{\text{refractory}} \tag{10}$$

## 4 RESULTS

### 4.1 EVALUATION WITH REAL-WORLD DATA

We first evaluate the DSD algorithm on real sEMG data acquired from 16 differential sensors worn around the circumference of the arm. The sEMG data recorded is shown in Figure 1. In this case, the human user was instructed to generate two separate activity modes, corresponding to a slight (imperceptible) tensing of muscles, first of the index finger and then of the small finger. Because our sEMG datasets lack simultaneously recorded intramuscular EMG to serve as ground truth signals (Negro et al., 2016), this evaluation is limited to unsupervised analyses.

Its subsequent blind factorization by the DSD is shown in Figure 2. We can clearly see the following facets: MUAPs are spatially localized with a clear spatial peak and fall-off. Their smoothness and time-profiles look reasonable, as expected for differential recordings, and the MUAPs look like they roughly factorize the data. Furthermore, their corresponding spike-trains also seem to spike at the appropriate peak locations. Lastly, the cardinality of the spike-trains/MUAPs looks reasonable as well, in this case parsimoniously explaining the dataset using two unique waveforms. The zoomed in versions of both results are shown for the orange and green waveforms in Figure 3. On the right hand side, we can see the reconstruction errors more saliently, and on the left hand side, we can see some missed detections corresponding to what look like low amplitude spikes.

### 4.2 QUANTITATIVE RESULTS WITH SIMULATED DATA

To quantify the DSD performance, we generate a family of simulated datasets using real-data (see Appendix C, similar to the hybrid ground truth approach used by Pachitariu et al. (2016) and Rossant et al. (2016). The main goal here is to compare MUAP waveforms similarities and corresponding

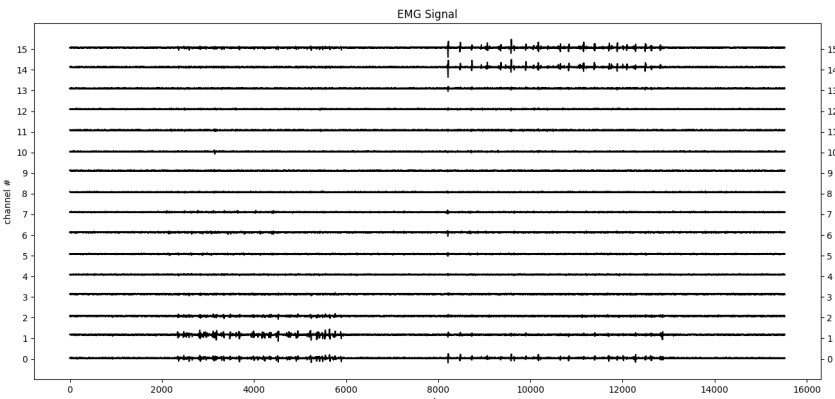

Figure 1: Recorded 16-channel wrist sEMG data with twitches in index finger in the first active section, and twitches in pinkie in the second active section.

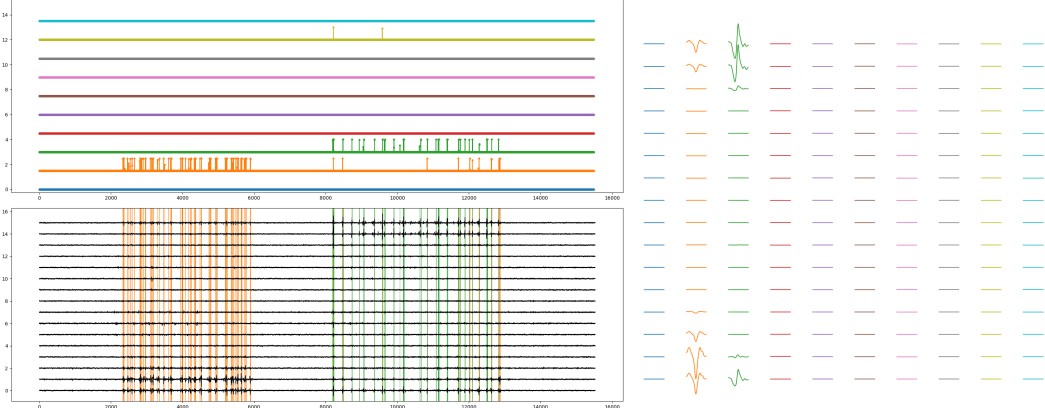

Figure 2: DSD-D factored sEMG data with 10 learned MUAP templates on the right, 10 corresponding spike trains on the upper left, and spike firing time overlaid on the sEMG on the lower left.

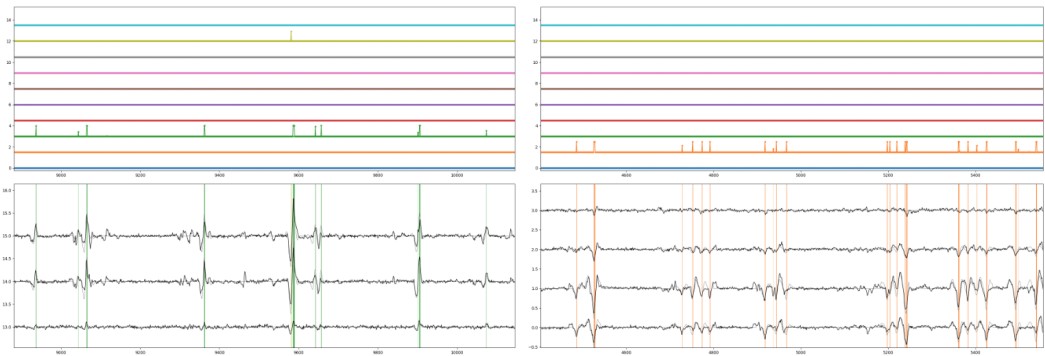

Figure 3: DSD-D factored sEMG data, zoomed-in sections of the spike trains and sEMG. On the left shows the pinkie active section, and on the right shows the index finger active section.

Table 1: sEMG decomposition evaluation results on simulated datasets

| Dataset | Metric | KiloSort | DSD-T | DSD-D |
|---|---|---|---|---|
| 3 MUs | recall | 0.6667 | **1.0000** | **1.0000** |
| | dissimilarity | 0.7112 | **0.0802** | 0.0937 |
| | cardinality_mismatch | -0.3333 | **0.0000** | **0.0000** |
| | overall_avg_prt | 0.6240 | **0.7899** | 0.7818 |
| | overall_avg_prt* | | **0.8089** | 0.7570 |
| 3 MUs | recall | 0.6667 | **1.0000** | **1.0000** |
| w/ high noise | dissimilarity | 0.6541 | **0.0802** | 0.1236 |
| | cardinality_mismatch | -0.3333 | **0.0000** | **0.0000** |
| | overall_avg_prt | 0.3532 | **0.8063** | 0.7428 |
| | overall_avg_prt* | | **0.8126** | 0.7434 |
| 5 MUs | recall | 0.6000 | **1.0000** | **1.0000** |
| | dissimilarity | 0.6068 | 0.3247 | **0.2255** |
| | cardinality_mismatch | **0.0000** | **0.0000** | 0.4000 |
| | overall_avg_prt | 0.5955 | 0.4539 | **0.7553** |
| | overall_avg_prt* | | 0.4835 | **0.6210** |
| 7 MUs | recall | 0.1429 | 0.8571 | **1.0000** |
| | dissimilarity | 0.7786 | **0.1484** | 0.2082 |
| | cardinality_mismatch | **0.0000** | **0.0000** | 0.1429 |
| | overall_avg_prt | 0.6452 | 0.5849 | **0.6535** |
| | overall_avg_prt* | | 0.5868 | **0.6537** |
| 9 MUs | recall | 0.4444 | **0.8889** | 0.7778 |
| | dissimilarity | 0.6012 | 0.1288 | **0.1188** |
| | cardinality_mismatch | -0.2222 | **-0.1111** | **-0.1111** |
| | overall_avg_prt | 0.4625 | **0.6112** | 0.5492 |
| | overall_avg_prt* | | **0.6174** | 0.5412 |

* indicates results on validation datasets for DSD

spike-train timing accuracies against this ground truth. Those are encoded in the following four metrics below:

1. **cardinality_mismatch:** The normalized cardinality difference between inferred and ground-truth MUAPs.
2. **recall:** The percentage of MUAPs successfully associated to ground-truth MUAPs.
3. **dissimilarity:** The reconstruction accuracy of the associated MUAPs.
4. **overall_avg_prt:** The timing accuracies of the inferred spike trains.

See Appendix B for a detailed description of these metrics and how they're calculated.

We evaluate two DSD variants, DSD-T, and DSD-D, against each other and also against the established spike sorting algorithm KiloSort1[1] (Pachitariu et al., 2016). DSD-T uses the tiramisu encoder and is trained with reconstruction, sparsity, parsimony, and refractory losses. Similarly, DSD-D uses simply replaces replaces the refractory loss with the uniqueness loss, and has the DenseNet encoder front end.

The DSD-T models were trained with a batch size of 32, using the Adam optimizer (Kingma & Ba, 2014) with an initial learning rate of 0.001, and the Gumbel-Softmax exponential temperature decay for every 32000 batches from max temperature, 1.5 to min temperature, 0.05. Dropouts were added in the tiramisu dense blocks, zeroing out channels with probability 0.3 during training. The DSD-D models were trained with a batch size of 16, using the Adam optimizer with an initial learning rate of 0.0006, and the Gumbel-Softmax inverse time constant of 0.0001 decreasing temperature every 1024 batches. Dropouts were added to the output of dense layers, zeroing out channels with probability 0.01 during training. Both were run using the Pytorch 1.0 framework (Paszke et al., 2017).

---

[1]KiloSort1 github repo: https://github.com/cortex-lab/KiloSort

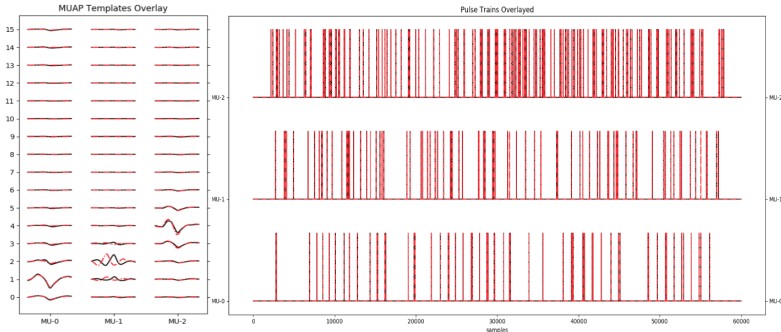

Figure 4: 3 MUAPs with high noise dataset. DSD-T results of inferred MUAP templates and spike trains (red) and ground-truth (black)
Note: The shifted MU-1 template doesn't affect the template score, but causes an offset in the corresponding spike train, which affects the overall_avg_prt score

For those experiments, the cardinality range of a dataset's underlying MUAPs are known, and sets of hyperparameters for both KS-1 and DSD-X were tuned accordingly. This was done so as to assess: i) How easy and meaningful hyperparameter tunings is for both algorithm families, and ii) to assess how well their performance is *even* if cardinality ranges are known ahead of time. Table 1 summarizes those results.

A visualization of the results corresponding to the second row of the Table 1 is shown in Figures 4 and 5. Qualitatively, we can see that the DSD's recovered MUAPs and their corresponding spike-trains are much more aligned with those of the ground truth. (Note the temporal shift of one of the MUAP's signatures) This does not affect MUAP scoring, as this temporal shift is accounted for in the shifted-Farina-distance (see Appendix B). This does cause a corresponding misalignment of the spike-train however, and this can manifest as a penalization in the overall_avg_prt score, as it won't be until a larger window is considered before which the offset spike is accounted for. The kilosort result appears to have MUAPs that are much more oversegmented overall, and one is missing altogether. The corresponding spike-trains also have have one that is also missing.

In general, we can see that the DSD results seem to be much more performant across the board with respect to KiloSort, (at least on those short 30 second datasets). The DSD appears able to recover a good amount of the underlying MUAP waveform shapes faithfully, and this appears to be fairly consistent across all datasets. In regards to the spike-train timing accuracies, the DSD family appears to also perform very well relative to KiloSort, although as the number of active MUAPs increase, both the KiloSort and DSD families appear to show a beginning trend towards becoming more or less similar in performance. On the dataset with 7 MUAPs for example, the DSD-D is just above KiloSort in performance on spike-train accuracies, and KiloSort beats out DSD-T.

As DSD-T and DSD-D were trained on different architectures and loss parameters, we can also gain insights into what good combinations are useful for which cases. Of important note, was the ease at which we were able to tune the hyperparameters of DSD in this context, owing to the interpretability of the loss terms. For example, avoiding MUAP shape splittings was readily tunable via the $L_{uniqueness}$ loss, (which this loss was specifically designed for). Similarly, tuning for excessive spike-trains was facilitated by increasing the $L1$. We view this interpretability of the DSD's loss functions as an advantage that comes "for free", since the DNN architecture incorporates biophysical constraints from the ground up.

All training sets were 30 seconds in length. The training time for KiloSort took 2 seconds, and ran on a Intel(R) Core(TM) i7-8750H CPU @ 2.20GHz. KiloSort does not have a concept of inference time, as it runs directly on a dataset and can immediately recover spike-trains and the corresponding MUAP waveforms. For the DSD, the training time could very anywhere from 10 minutes to 1 hour, depending on the number of epochs utilized. Training the DSD took place on one nVidia GeForce GTX 1070 GPU. Inference time of the DSD, which also took place on the GPU, was $< 1$ ms.

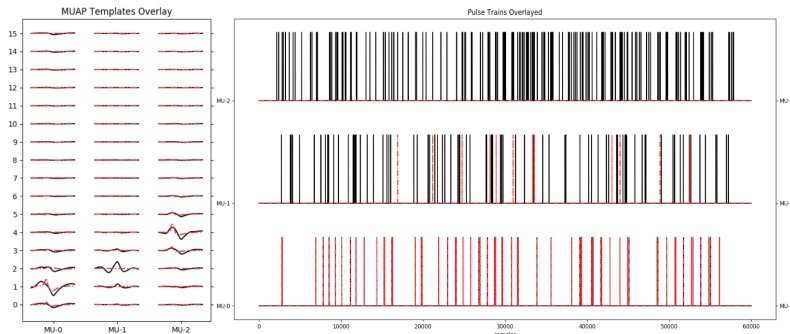

Figure 5: 3 MUAPs with high noise dataset. KiloSort results of inferred MUAP templates and spike trains (red) and ground-truth (black)

## 5  CONCLUSION

We present a novel self-supervised learning algorithm for spike sorting of multichannel sEMG data. This model marries the power of deep end-to-end neural-networks with the inductive biases of bio-physical models. The architectural novelty is forcing the deep net to reconstruct sEMG through the bottleneck of the biophysical model that created it in the first place, while the loss novelties are around the explainability and interpretability of them vis-a-vis the biophysics. The architecture therefore forces the net to learn the appropriate generative latent variables - the spike-trains, and their corresponding MUAP waveforms, and the loss terms provide clear physical meaning for tuning the model. The algorithm is elegantly self-supervised, and requires no training labels whatsoever. The model once trained, can then be applied as is to similar data sets to provide spike-extraction. Relative to an established spike-sorting method, this autoencoder-based approach shows superior recovery of source waveforms and event times at various degrees of superposition, (at least on short duration data sets).

The DSD framework opens up a plethora of exciting avenues for the application of self-supervision to complex signals such as sEMG. More broadly however, it showcases an exciting and concrete methodology for incorporating physics-based priors hand in hand with the power of end-to-end deep learning.

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

## A    GUMBEL-SOFTMAX BINARIZATION OF SPIKE PULSE TRAINS

The spike probabilities predicted by the DSD encoder are binarized using the Gumbel-softmax categorization trick in order to avoid partial spikes. This operation is given in Equation 11.

$$P(z_i) = \frac{e^{\frac{log(P(y_i)) + g_i}{\tau}}}{\sum_{j=1}^{C} e^{\frac{log(P(y_j)) + g_j}{\tau}}} \tag{11}$$

It is similar to the standard softmax operation with two new terms are added: the temperature $\tau$ parameter and a draw from a Gumbel distribution $g \sim G(0,1)$. The Gumbel draw can be computed by $g = -log(-log(u))$ where $u \sim U(0,1)$ is a draw from a uniform distribution.

During training time, as the temperature term $\tau$ is reduced, the distribution of spike probabilities begins to approximate a one-hot vector, forcing the DNN to make spike v.s. non-spike decisions. During inference time, we drop the stochastic element altogether and round the resulting probabilities.

## B    EVALUATION FRAMEWORK

The evaluation framework evaluates the correctness of MUAP waveforms and Spike Trains recovered by spike decomposition algorithms. It does so by comparing ground truth and recovered MUAP waveforms, and also comparing ground truth and recovered spike trains.

### B.1    EVALUATING MUAP WAVEFORM RECOVERY

MUAP Waveform recovery is evaluated by first associating each ground truth MUAP waveform to a recovered MUAP waveform and then computing metrics, as described in the following sections.

#### B.1.1    MUAP WAVEFORM ASSOCIATION

Each ground truth MUAP waveform is associated to the best matching recovered MUAP waveform. Given the Normalized shifted Farina distance (sfd) metric, as in Merletti & Farina (2016), between pairs of ground truth and recovered MUAP waveforms, matches are found using the Hungarian algorithm (Kuhn, 1955).

sfd is computed as follows:

$$\text{normalize(mu)} = \frac{\text{mu} - \mu}{\|\text{mu}\|_2} \tag{12}$$

$$\text{energy(mu)} = \|\text{mu}\|_2^2 \tag{13}$$

$$\text{sfd(mu1, mu2)} = \frac{2 * \text{energy}(\hat{\text{mu}}1 - \hat{\text{mu}}2\_\text{aligned})}{\text{energy}(\hat{\text{mu}}1) + \text{energy}(\hat{\text{mu}}2\_\text{aligned})} \tag{14}$$

where $\mu$ is the per channel mean of the corresponding MUAP, mu1 and mu2 are normalized as above to yield $\hat{\text{mu}}1$ and $\hat{\text{mu}}2$, $\hat{\text{mu}}2$ is temporally aligned to $\hat{\text{mu}}1$ to yield $\hat{\text{mu}}2\_\text{aligned}$ such that their correlation is maximum.

#### B.1.2    RECALL

Recall is computed as the fraction of MUAP waveforms recovered correctly. A value of 1 is the best score and a value of 0 is the worst.

$$\text{recall} = \frac{N_{\text{correct\_recoveries}}}{N_{\text{gt\_muaps}}} \tag{15}$$

Here, $N_{\text{correct\_recoveries}}$ is computed as number of instances where sfd between a ground truth and it's associated MUAP waveform is below a threshold. (default: 0.9). $N_{\text{gt\_muaps}}$ is the number of motor units in the ground truth.

### B.1.3 CARDINALITY MISMATCH

Cardinality mismatch is computed as the difference in number of recovered and ground truth MUAP waveforms normalized by the number of ground truth waveforms.

$$\text{cardinality\_mismatch} = \frac{N_{\text{recovered\_muaps}} - N_{\text{gt\_muaps}}}{N_{\text{gt\_muaps}}} \tag{16}$$

This measure reflects scenarios where the algorithm failed to recover the exact number of ground truth MUAP waveforms. A value of 0 is the best score. A value $< 0$ indicates that we failed to recover some MUAP waveforms, and a value $> 0$ indicates that we recovered more MUAP waveforms than ground truth.

### B.1.4 DISSIMILARITY

For all correct recoveries the dissimilarity is computed as the average sfd distance between ground truth MUAP waveforms and it's associated MUAP waveforms.

$$\text{dissimilarity} = \frac{\sum_{i=1}^{N_{\text{correct\_recoveries}}} \text{sfd}\left(\text{gt\_mu}_i, \text{associated\_mu}_i\right)}{N_{\text{correct\_recoveries}}} \tag{17}$$

This measure indicates how accurately recovered MUAP waveforms were reconstructed. A value of 0 is the best score, anything higher is progressively worse.

## B.2 EVALUATING SPIKE TRAIN RECOVERY

Spike Train recovery is evaluated by first associating each ground truth Spike Train to a recovered Spike Train and then computing metrics, as described in the following sections.

### B.2.1 SPIKE TRAIN ASSOCIATION

Each ground truth spike train needs to be associated to the best matching recovered spike train. Ground truth spike trains are binary. The recovered spike-trains are probabilities.

Let us assume that there were no alignment issues between ground truth and recovered spike trains. In such a case, one could simply measure the L1 distance between ground truth and recovered spike trains, and come up with the best association.

Realistically however, alignment issues between spike trains can exist, even though they may have the same inherent firing pattern.

Therefore we modify the association measure, to incorporate mis-alignments. We do this in the following way: The L1 measure is first applied to all samples across ground truth and recovered spike trains, except for all regions around ground truth *spikes*.

Those regions, are defined by +/- 20 samples around any given ground truth spike. Within this region, we simply pick the max spike probability from the recovered spike train, and use this value, we ignore all other values within the region.

For every region, we now have a list of ground truth spikes and a list of max probability values corresponding to the ground truth spikes. We simply compute the L1 measure of the two lists and sum it to the L1 measure computed above. This is the associated spikes distance (asd).

Given the asd metric between pairs of ground truth and recovered spike trains, matching spike trains are found using the Hungarian algorithm (Kuhn, 1955).

### B.2.2 OVERALL AVERAGE PRECISION-RECALL-TOLERANCE

Each ground truth spike train needs to be evaluated against the best matching recovered spike train. Ground truth spike trains are binary. The recovered spike-trains are probabilities.

Let us assume there were no alignment issues between ground truth and recovered spike trains. In such a case, one could measure algorithm performance by computing a precision-recall curve for a ground truth and its associated spike train.

The average precision (ap) score computed from the precision-recall curve above could be used to measure recovery performance for a single spike train.

However, alignment issues between spike trains can exist. Therefore we modify the evaluation measure, to incorporate mis-alignments.

Let us consider regions, defined by +/- samples tolerance window around a ground truth *spike*, we detect the max spike probability from the recovered spike train in these regions. We temporally align the max spike probability with the ground truth spike in that region. All the other values within this region are left as is.

We now have aligned spike trains, for which we can calculate a single precision-recall curve and its average precision score.

We can repeat this process for a range of tolerance window values, to obtain a bank of precision recall curves and a list of average precision scores. This accommodates a range of mis-alignment errors.

We can average all the average precision scores to obtain the precision-recall-tolerance (prt) score for one spike train. This score measures the accuracy of spike train recovery.

Overall average precision-recall-tolerance is simply the average of all prt scores for all spike trains. This is a measure of how accurately we can recover all the spike trains from sEMG data. A value of 1 is the best score and a value of 0 is the worst.

$$\text{average\_prt} = \frac{\sum_{i=1}^{N_{\text{gt\_sts}}} \text{prt}(\text{gt\_st}_i, \text{associated\_st}_i)}{N_{\text{gt\_sts}}} \tag{18}$$

$$\text{prt}(\text{st}_1, \text{st}_2) = \frac{\sum_{win=0}^{10ms} \text{ap}(\text{st}_1, \text{st}_2, \text{win})}{20} \tag{19}$$

We pick a maximum tolerance window (win) of 10ms around a ground truth spike to represent half the minimum interval between consecutive spikes firing from the same motor unit in the forearm. We increment win sizes by 0.5ms.

## C  SIMULATED DATASET GENERATION

The simulated datasets are 30 seconds long, 60,000 samples in length, with 16 channels in width. They are generated using the extracted MUAP templates and spike trains from real-world sEMG datasets. It is a human involved extraction process to ensure the quality of the spike trains and MUAP templates. After we have a bank of MUAP templates and spike trains, we select several ones we want to use in a dataset. The dataset generation process is convolution of the MUAP templates with the spike trains and then adding Gaussian noise. In this way, the simulated datasets closly resemble the real-world sEMG datasets, but with the ground-truth spike trains and MU templates for quantitative evaluation. We generate these datasets with different number of MUs and superposition in order to vary the spike sorting difficulty level and thus test the algorithms thoroughly.

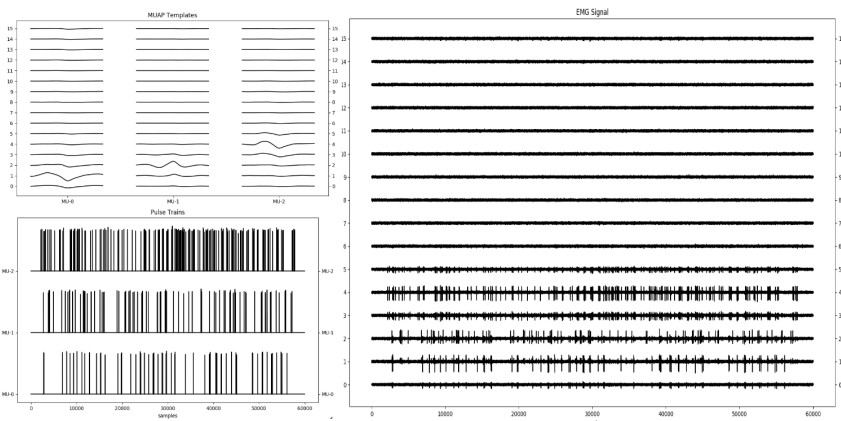

Figure 6: Simulated dataset with ground-truth MUAP templates and spike trains

More specifically, there are 5 simulated datasets we use in the evaluation. The 3 MU dataset has 3 MU templates with one channel offset which is the most simple and non-trivial case. The 3 MU with high noise dataset has the same MUs as the 3 MU dataset except the noise power is increased by 4 times, which is to test the performance on lower SNR datasets. The 5 MU dataset has 3 MU templates on the same channel and 2 MU templates on another channel, which is to test the capability of addressing the spatial superposition problem. The 7 MU dataset has 5 MU templates with one channel offset, and 2 MU templates on the same channel. The 9 MU dataset has 5 MU templates with one channel offset and 4 MU templates on the same channel. The last two datasets are the stress-test cases by increasing the amount of MUs and spatial superpostion. Figure 6 shows the example of 3 MU simulated dataset on the right, the MUAP templates used on the upper left, and the spike trains used on the lower left.

