# OpenReview forum: "Deep Spike Decoder (DSD)"
_ICLR.cc/2020/Conference — Reject_

### Official Review · AnonReviewer1 · 2019-10-26
**Official Blind Review #1**

**Rating:** 1

**Review:**

This paper describes a machine learning model for learning spiking representations from EMG (electromyography) like data. The basic model is an autoencoder, where the bottleneck layer is designed to project the waveform inputs into "spike" representations. Specifically, there is a DSD (Deep Spike Decoder) encoder and decoder, where the encoder is implemented as either a DenseNet[1]-like or Tiramisu[2]-like architecture. The bottleneck seems to be implemented by a linear layer binarized with a Gumbel-Softmax activation. The decoder is a linear layer. Several losses are considered, including a L4-norm (!), a sparsity loss, a parsimony loss, a uniqueness loss, and a refractory period loss. The model is validated qualitatively on real EMGs, and quantitatively on synthetic data.

The strengths of the paper are the following:
- Spiking models are very interesting class of models, which if attained would have a great impact on several other areas of machine learning.

- I like the straightforward design that is fit to the purpose of generating spiking representations. A Gumbel-Softmax has proven its validity and is a logical fit to the problem setup.

- I like the simpler setup the paper chooses.
  -- First, the paper chooses not to take the route of trying to learn backpropagation through infinite time steps, as often happens in spiking methods. This is a beast on each own, which would be nice to set as an ultimate goal (in my opinion). For now, however, it's ok if we forego this requirement.
  -- Second, the paper chooses not to assume that signals come asynchronously, which again makes things unnecessarily complex, given the state of the field.

The weaknesses of the paper:
- While the problem setup is simple, perhaps it makes some assumptions that are too strong. In my opinion, the strongest one is that of batch-learning (for the lack of a better name). Often, spiking models are studied in an online setup where the data comes online (continuously), and then spikes are generated when sufficient changes in the inputs warrant a delta-difference (spike). When in batch mode, however, it should be quite much easier to obtain spikes that lead to good reconstructions, as there is no much need for a "memory" mechanism. While one could argue whether an online setup is necessary or not, in my opinion it is necessary and would make a spiking model challenging and interesting to learn. Otherwise, it looks like a conventional autoencoder, only with spikes instead of dense representations.

- The model is unclear and writing vague.

- First of all, after reading the abstract and the introduction I get the feeling that the model is probabilistic, as there is mention of priors and autoencoders. Also, later on a Gumbel-Softax is mentioned for binarization. Gumbel-Softmax is a continuous function and binarization makes sense when sampling, that is when assuming a generative process. However, the rest of the paper seems not to explain a probabilistic or generative model. There is no explicit prior discussed. There is no sampling procedure discussed. The losses are explained but not within a particular probabilistic or stochastic framework. If the model is not stochastic, one way or the other, how are the discrete spikes obtained and how is the system is trained?

- All the loss functions, albeit logical when taken one by one, they do look ad hoc and appearing out of the blue. This perhaps relates to the previous point, where it is not clear if the model is stochastic or deterministic. If it is deterministic, it is ok to have all the loss functions appearing like that, but I would expect some more explanation on what purpose do they fill w.r.t. the spiking model. In the end, what does the deep spiker model try to achieve? Learn spikes as representations? Recover the original spikes? Be sparse? If yes, why and how sparse? Be energy-efficient?

- Some design choices are quite unclear. Generally, it is fair to say that the experimental and design setups are rather simple: multiple 1-D waveforms and not much noise (from what I get). In that context, it is not clear why DenseNets, or even Tiramisu-Nets are used as an encoder; especially when the decoder is a simple linear model.

- Also, phrases like "what good combinations are useful ... the hyperparametres of DSD" do not add to the clarity. There is no exploration of hyperparameters in the experiments and no individual examination of the contribution of each loss (unless I missed it somewhere).

- Similarly, what does "For the DSD ... between 10 minutes to an hour depending on ...".  Such statements should be more precise, for instance plotting wall clock time vs training loss.

All in all, while I like the motivation and the original direction, I believe there exist a lot of questions  unanswered before acceptance.

**Experience Assessment:**

I have published in this field for several years.

**Review Assessment: Checking Correctness Of Derivations And Theory:**

I carefully checked the derivations and theory.

**Review Assessment: Checking Correctness Of Experiments:**

I carefully checked the experiments.

**Review Assessment: Thoroughness In Paper Reading:**

I read the paper at least twice and used my best judgement in assessing the paper.

---

### Official Review · AnonReviewer4 · 2019-11-06
**Official Blind Review #4**

**Rating:** 1

**Review:**

This paper proposes a new algorithm for spike-sorting. It is implemented by a deep autoencoder with biophysically motivated loss functions. The encoder is the main module which conducts the spike-sorting task while the decoder is only used for training the model in an end-to-end fashion.

This paper should be rejected due to the following arguments:
- The paper lacks a section on literature survey, to let the reader know how/where the proposed method fills the gap in the current state-of-the-art. They do compare their results with the KiloSort (Pachitariu et al., 2016) algorithms, however, no discussion is provided on how it works and why their method outperforms it.
- It is unclear why the reconstruction loss is chosen to be an L4 norm as opposed to L2.
- The authors claim that the parsimony loss as defined in Eq. (7) forces “the network to be parsimonious with respect to the set of MUAP proposals it uses to explain a given sEMG signal.” My understanding, however, is that the only functionality of the loss defined in Eq. (7) is to enforce temporal smoothness. More elaborate explanation is needed to support the authors claim.
- I could not understand the functionality of the uniqueness loss. Specifically, why should “the joint occurrence of the temporal similarity between MU spike trains and the spatial similarity between their MUAP waveforms” be penalized? Isn’t that the case that same stimuli should result in similar response? It is unclear what this has to do with forcing to explain different phenomena.

Things to improve the paper that did not impact the score:
- The method (and the paper) is named deep spike “decoder” (DSD) while in fact the “encoder” part of the learned deep autoencoder actually conducts the spike-sorting task. This could be confusing!
- Page 2, Sec. 3.1, line 2: Should use \times in inline equations in Latex for the multiplication symbol, not character x. Fix everywhere in the text.
- Page 6, Par. -2, line -2: The word “replicate” is repeated.
- Non-legible plots axes.



**Experience Assessment:**

I do not know much about this area.

**Review Assessment: Checking Correctness Of Derivations And Theory:**

I carefully checked the derivations and theory.

**Review Assessment: Checking Correctness Of Experiments:**

I assessed the sensibility of the experiments.

**Review Assessment: Thoroughness In Paper Reading:**

I read the paper at least twice and used my best judgement in assessing the paper.

---

> ### Author Response · Authors · 2019-11-08
> **Discussion of Review #4**
>
> Thanks a lot for the review. In response to your points:
>
> - We avoided extensive literature review and explanation on KiloSort mostly due to page limitations. In the future revised version of the paper, we’ll try to provide more background information and also include a summary of KiloSort in the appendix.
>
> - We chose L4 over L2 largely due to ease of hyperparameter tuning purposes. EMG signal contains both action potentials and noise. Sparsity and reconstruction loss terms, by creating a trade-off, are tuned such that only the signal corresponding to action potentials are reconstructed. Here, we use the L4 norm because it differentiates between action potential signals from EMG noise more successfully and makes the hyperparameter tuning (see total loss equation) easier.
>
> - Parsimony loss is based on the L1-L2 norm. The L1-L2 norm (and more generally the L1-Lq norm) is used as a block sparsity inducing penalty in optimization algorithms (see [1]). The critical part in the parsimony loss function is the tensor partitioning G over which L1 norm (i.e. the sum operation) is performed. By applying the L1-L2 norm over different partitionings of a tensor, one can induce different patterns of block sparsity. Here, our partitioning is based on spatial neighborhoods (specified by the number of consecutive electrodes) for each individual motor unit. This loss then minimizes both the number of motor units and their spatial footprints. The reason why we apply this loss on the first time-derivative of the spatiotemporal waveforms tensor is to get the temporal smoothness effect as well. We’ll try to complement our explanation with visualizations in the revised paper.
>
> - The uniqueness loss (and also refractory period loss) are optional terms we use to tackle particular problems we observed during our experiments. We’ll move the experimentations with these terms to the experiments section and explain them in relation to the particular problems they’re trying to address.
>
> [1] Francis Bach, Rodolphe Jenatton, Julien Mairal, Guillaume Obozinski, et al. Optimization with sparsity-inducing penalties. Foundations and Trends in Machine Learning, 4(1):1–106, 2012.

---

### Decision · Program_Chairs · 2019-12-19

**Decision:**

Reject

**Comment:**

The paper presents a model for learning spiking representations. The basic model is a a deep autoencoder trained end-to-end with a biophysical generative model and results are presented on EMG and sEMG data, with the aim to motivate further research in self-supervised learning.

The reviewers raised several points about the paper. Reviewer 1 raised concerns about lack of context on surrounding work, clarity of the model itself and motivating the loss. Reviewer 2 pointed out strengths of the paper in its simplicity and the importance of this problem, but also raised concerns about the papers clarity, again motivations on the loss function and sensibility of design choices. The authors responded to the feedback from reviewer 1, but overall the reviewer did not think their scores should be changed.

The paper in its current form is not yet ready for acceptance, and we hope there has been useful feedback from the reviewing process for their future research.